



# Using geomorphometry for hydro-geomorphological analysis in a Mediterranean research catchment.

Domenico Guida[1], Albina Cuomo[1], Vincenzo Palmieri[2]

[1]Department of Civil Engineering, University of Salerno, Fisciano, 84084, Italy
[2]ARCADIS, Agency for Soil Defense of the Campania Region, Naples, Italy

*Correspondence to*: A. Cuomo (acuomo@unisa.it)

**Abstract.** The aim of the paper is to apply an object-based geomorphometric procedure to define the runoff contribution areas and support a hydro-geomorphological analysis on a 3-km$^2$ Mediterranean research catchment (southern Italy). Daily and sub-hourly discharge and electrical conductivity data were collected and recorded based on three-year monitoring activity. Hydro-chemograph analyses on these data revealed a strong seasonal hydrological response in the catchment that were different from the stormflow events that occurred in the wet period and in dry periods. This analysis enabled us to define the hydro-chemograph signatures related to increasing flood magnitude, which progressively involves various runoff components (base flow, subsurface flow and surficial flow) and an increasing contributing area to discharge. Field surveys and water table/discharge measurements carried out during a selected storm event enabled us to identify and map specific runoff source areas with homogeneous geomorphological units previously defined as hydro-geomorpho-types (spring points, diffuse seepage along the main channel, seepage along the riparian corridors, diffuse outflow from hillslope taluses and concentrate sapping from colluvial hollows). Following the procedures previously proposed and used by authors for object-based geomorphological mapping, a hydro-geomorphologically-oriented segmentation and classification was performed with an e-Cognition (Trimble, Inc) package. The best agreement with the expert-based geomorphological mapping was obtained with weighted profile and plane curvature sum at different-size windows. Combining the hydro-chemical analysis and object-based hydro-geomorpho-type map, the variability of the contribution areas was graphically modelled for the selected event which occurred during the wet season by using the log values of flow accumulation that better fit the contribution areas. The results enabled us to identify the runoff component on hydro-chemograph for each time step and to calculate a specific discharge contribution from each hydro-geomorpho-type. This kind of approach could be useful applied to similar, rainfall-dominated, forested and no-karst catchments in the Mediterranean eco-region.

Keywords: geomorphometry, hydro-geomorphology, runoff contributing area, Cilento Global Geopark

## 1 Introduction

In order to gain a better understanding of hydrology, it is essential to study the complex interactions and linkages between watershed components, such as drainage network, riparian corridors, headwaters, hillslopes and aquifers and related processes operating at multiple scales (National Research Council, 1999). Hydrological science plays an important and fundamental role only when it provides an integrated knowledge and understanding of the forms and processes that operate in watershed at multiple, space-time scales in the landscape (Marcus et al., 2004). A useful way of understanding the response of catchments to rainfall events is to analyze stream discharge vs rainfall per unit of time, plotted as a storm flow hydrograph and hyetograph, respectively. In recent decades, hydrologists have carried out numerous studies on catchment and hillslope hydrology in order to define when, how and where runoff is produced and how it progressively increases along the drainage network. Hydrologists generally agree that following rainfall, new-event water components are added, through various hydrological mechanisms to





the old, pre-event water components which are generally referred to as base flow components that derive from deep and shallow aquifers, expanding and reducing the runoff-contributing areas (Betson, 1964). The most common general concept that explains the above-mentioned hysteretic behavior is the Variable Source Area (VSA) concept. This concept was originally proposed by Hewlett (1961) and later adopted by other authors (Dunne and Black, 1970; Dunne and Leopold, 1978, Huang and Laften 1996, Vander Kwaak and Loague 2001, Zollweg et al. 1995, Pionke et al. 1996). Despite its early formulation, it represented the hydrological background for more recent research studies (Lyon et al. 2004, Easton et al. 2007, 2008, Buchanana et al. 2012, Moore et al. 1988, Barling et al. 1994, Kwaad 1991, Easton et al. 2010, White et al. 2011). Contemporarily, the "hydro-geomorphic paradigm" was proposed by Sidle et al. (2000) in order to discriminate the VSA hydrologic sources and pathways, which refers to the connected hydro-geomorphic components of the catchments (hollow, hillslope and riparian corridor). Within a more general program for flood hazard assessment procedures, the hydro-geomorphic paradigm will be used to generalize at basin and regional scale in southern Italy by Cuomo (2012), by means of hydro-geomorphology (Okunishi, 1991; Okunishi, K., 1994; Babar, 2005; Sidle and Onda, 2004; Goerl, Kobiyama, dos Santos , 2012). Cuomo (2012) introduced and applied a new hydro-geomorphological basic unit: the *hydro-geomorpho-type*, by using the Salerno Geomorphological Mapping System (Dramis et al., 2011; Guida et al., 2012; Guida et al., 2015),for object-based geomorphological mapping. This proposal is currently under experimental calibration as an effective, object-based geo-morphometry procedure for spatial individuation, objective delimitation and automatic recognition of the hydro-geomorpho-types, in the perspective of an object-based distributed hydrological modelling (Cuomo et al., 2012).

Linking geomorphometry with hydrology toward the hydro-geomorphology gives consistency to the suggestion made by Peckam (2011) with the aim of simplifying the issue of the computational cost and time of a fully distributed model.

In the past, many authors made extensive use of chemical and isotopic tracers in order to separate the runoff components recorded in the hydrographs and pinpoint distinctive sources and pathways by using the geochemical and isotopic signature of water at parcel scale or for small catchments (Klaus and McDonnell, 2013). However, applying only the hydro-chemograph and isotopic separation methods to an experimental parcel cannot provide sufficient information on the spatial distribution of runoff sources and paths for basins as a whole, due to their spatial heterogeneity structure and time process variability.

Moreover, extensive use of the above-mentioned methods is more expensive and time-consuming than the quantity and quality of the data collected and the knowledge gained. As stated by Ladouche et al. (2001), with these methods alone it is possible to identify type, timing and volume of the runoff components, but it is impossible to define the spatial origin and related pathways during storm events accurately. In order to overcome these difficulties and by following the general approach used by Latron and Gallart (2007), we used an integrated, hydro-geomorphological approach for studying a Mediterranean research catchment in southern Italy. This approach is based on detailed geomorphological surveys, mapping and three-year hydro-chemical monitoring. It integrates a new procedure for identifying and separating hydro-chemical runoff components and a geomorphometric application for the objective delimitation of the source areas, where each runoff component is generated (Cuomo and Guida, 2013, Guida and Cuomo, 2014). Starting from these premises, the paper describes the study area as a Mediterranean research catchment and presents the hydro-chemical dataset recorded during the monitoring activity carried out in the 2013-2014 calibration period. In the next section an original procedure is explained for discriminating timing, type and hydro-chemical signature of the runoff components involved during storm events. With the aim of spatially defining these runoff sources, an object-based hydro-geomorphological map was then set by a hydrological-oriented segmentation and classification. Finally, the results of combined hydro-chemical and object-based hydro-geomorphometric analysis are discussed in order to determine the variability of the contribution area during a significant storm event.





## 2 Hydro-geomorphology and monitoring activity of the study area

The study area is a forested and hilly catchment located in the Bussento River drainage basin, the 3Km$^2$ Ciciriello catchment, located in the Cilento and Vallo di Diano National Park-UNESCO Global Geopark, Southern Italy (Fig. 1).

At the base the terrigenous bedrock is composed of a lower Tertiary, marly-clayey formation passing in unconformity upward

to middle Miocene, a westward-dipping sandstone strata and pelitic intervals. A lenticular 10 m thick marly layer ("Fogliarina Marl" geosite) outcrops along the right hand side of the valley. Regosols, regolite and gravelly slope deposits up to 5 meters thick, cover the bedrock mentioned above. The mainstream bed, rectilinear and dipping strata subsequent to main faults  is incised in alluvial gravelly and smooth deposits and partly in bedrock; the secondary streambed is exclusively in bedrock, subsequent to minor fault systems. From a hydro-geomorphological perspective, the groundwater circulation is controlled by

the litho-structural arrangement of the above-mentioned bedrock formations, where the marly-clayey formation constitutes the local aquitard below the sandstone aquifer. The westward dipping of the permeability boundary causes a general westward groundwater flow, convergent toward the lower apex of the wedge-like hydro-structures ("*hydro-wedge*" in Cascini at al., 2008), where the main permanent springs are located. In the headwaters, colluvial hollows are situated at the bottom of the zero-order basins, and are considered to be the main headwater hydro-geomorphotypes by Cuomo (2012), where dominant

saturation excess runoff occurs mainly during the wet season. The stream flow of both permanent springs from the bedrock aquifers and seasonal springs from colluvial headwater increase down valley.

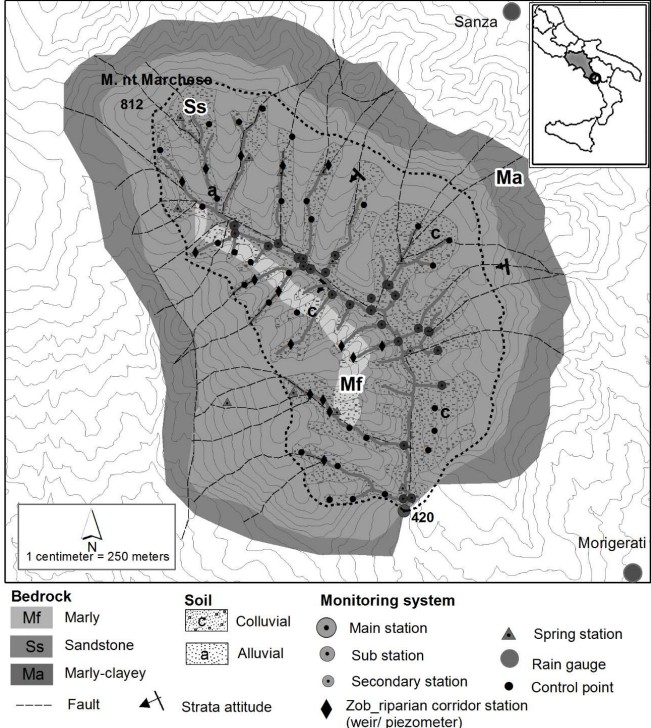

Figure 1. Location and geomorphological map of the Ciciriello Experimental Catchment (from Cuomo and Guida, 2016 under revision). Legend: Bedrock lithology; Ma: Marly-clayey and argillitic tertiary formation, base aquiclude; Ss: Sandstone

Miocene formation, fractured general aquifer; Mf: Marly limestone, interlayered and perched aquifer.





Since December 2012, water depth (D), discharge (Q) and electrical conductivity (EC) were measured daily at the main station, hourly during the floods and weekly at the sub-stations during the inter-storm periods (Fig. 1). The Q measurements were obtained with the Swoffer 3000 current meter (Swoffer Inc., USA), and the EC parameter was measured with the multi-parametric probe HI9828 (Hanna Instruments Inc., Romania). The monitoring year 2013-2014 (Fig. 2) provided a complete hydro-chemical dataset , which enabled us to carry out the analysis at seasonal and event time scales (Cuomo and Guida 2014).

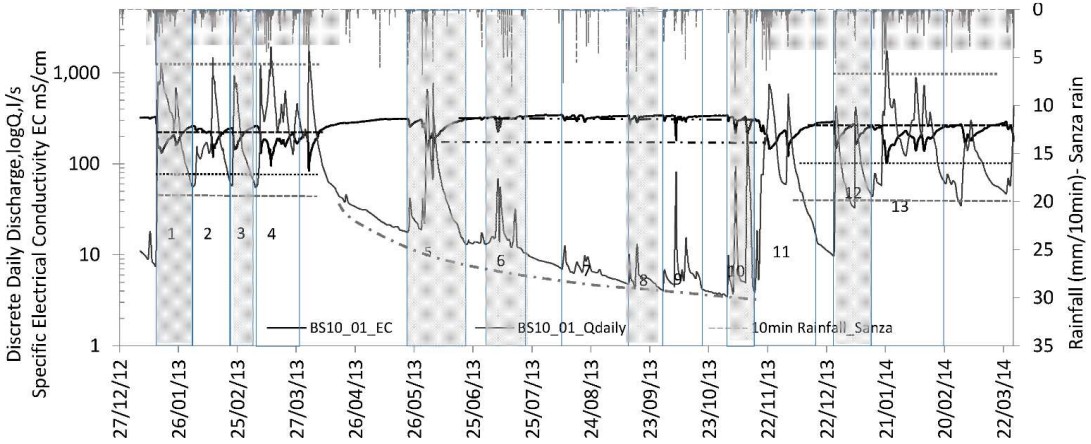

Figure 2. Plot of the hydro-chemograph dataset recorded at the main monitoring station (BS16_01) and the 10-min rainfall plot at the Sanza rain gauge (from Guida and Cuomo, 2016 under revision). Legend: Numbers indicate the selected events; horizontal lines are representative of the reference parameter ranges; black dashed-double dot lines indicate EC maxima in the dry period; black dashed-dot line represents EC minimum during the dry period; black dashed line indicates EC maximua in the wet period; black dotted line represents EC minimum in the wet period; gray dashed line indicates the Q minima in the wet period; gray dotted line indicates the average Q maximum in the wet period; finally, the gray dashed-dot curve indicates the theoretical annual base flow curve of the catchment during the period under consideration.

## 3. Hydro-geomorphological procedure for the contribution areas individuation

The contributing area is a dynamic hydrological concept because it may vary seasonally. The extension of the contributing area is strongly influenced by various static factors such as topography and soils, and dynamic factors such as antecedent moisture conditions, rainfall characteristics (Dunne and Black, 1975).

In the following sections, an integrate procedure is proposed that uses simple geomorphometric tools to take into account various hydrological and geomorphological factors which cause runoff variability on the catchment case study.

The flow chart in Fig. 3 shows the three integrated approaches used in the application.





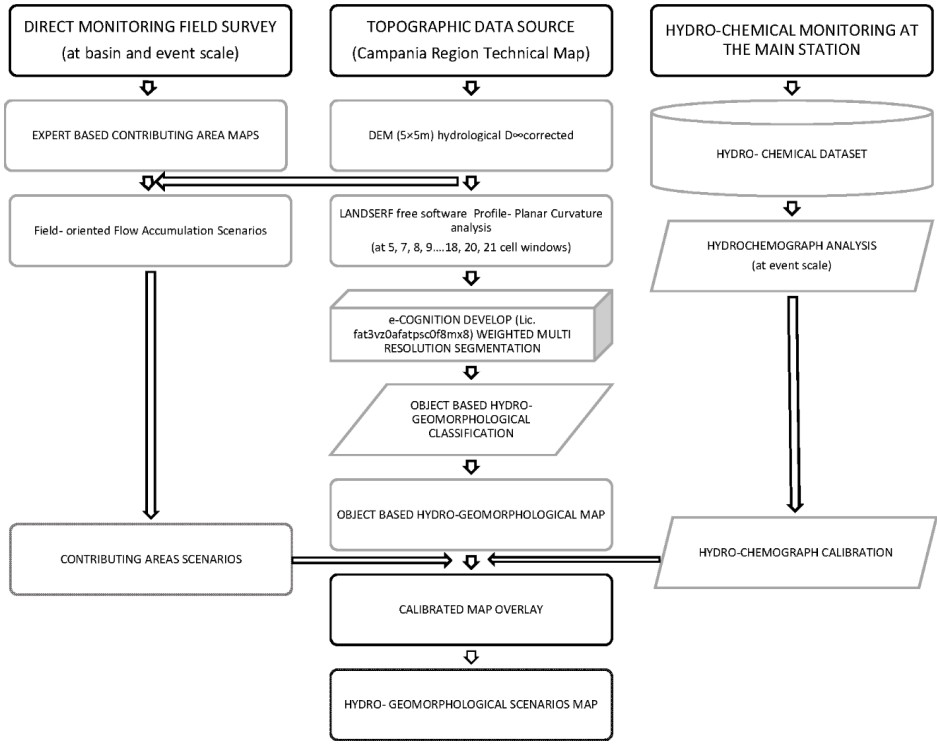

Figure 3. Flow chart procedure for identifying Contribution Areas

The first approach on the left hand side highlights the expert-based activities by direct monitoring carried out at basin scale
during the application event and the derivation of traditional, hand-draw, expert-based contribution area maps. The maps were
obtained by linking Q and sEC data collected at the control points (Fig. 1) for each event time step. The expert-based activities
are illustrated in Sect. 3.1. The second approach (see the flow at the center) shows the geomorphometric routine activities
performed during the application, as illustrated in Sect. 3.2. Starting from the topographic data source (Technical of the
Campania region), a hydrologically-corrected DEM was obtained by means of the D∞ algorithm and the best agreement
between the above expert-based maps and the flow accumulation maps enabled us to obtain the field-oriented accumulation
maps, as a proxy for the five contributing area scenarios. As better explained in Sect. 3.2, after five steps of elaborations, the
geomorphometric analysis provided us with the Object-based Hydro-geomorphological map of the catchment, quantitatively
defining spatial extension of the basic hydro-geomorphotypes. The hydro-geomorphotype map, was calibrated by the hydro-
chemical analysis illustrated in Sect. 3.3 and was then overlaid with the five contributing area scenarios thus obtaining the
final hydro-geomorphological scenarios maps.

## 3.1 Direct survey on the catchment during a storm event

Before and during the storm event in the period from 29 to 31 Jan 2015, one of the authors carried out direct field surveys by
measuring EC and, wherever possible, Q parameters on the control points in Fig. 1, and repeated them at each time-step of the
storm event. The pre-event conditions were detected at 5:15 pm on 29 Jan 2015 by carrying out systematic surveys and taking

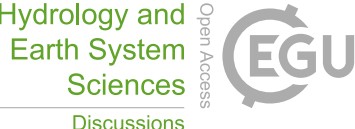



measurements from the main stream and secondary channel stations (Fig. 4a), where only groundwater feeds the discharge along the riparian corridors. After the beginning of rainfall, measurements were taken from 7:20 am to 9:10 am on 30 Jan 2015 at the zob springs and hollow stations (Fig.4b ), where the soils became more and more saturated and contemporarily new water was added from the riparian corridor downstream

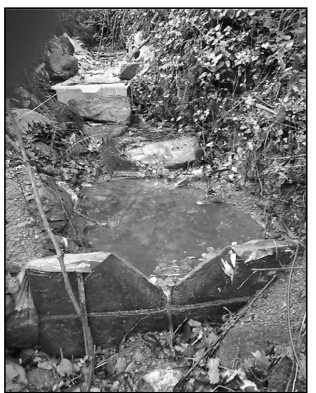

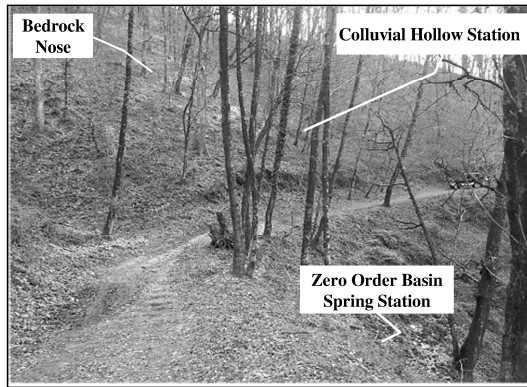

Figure 4. The V-notch weir at the BS16_01_01dx end station.

At the storm event peak, repeated measurements at the same control points were taken from 11:30 to 1:00 pm, which detected direct runoff ( Fig. 5a) and soil pipe (fig. 5b) .

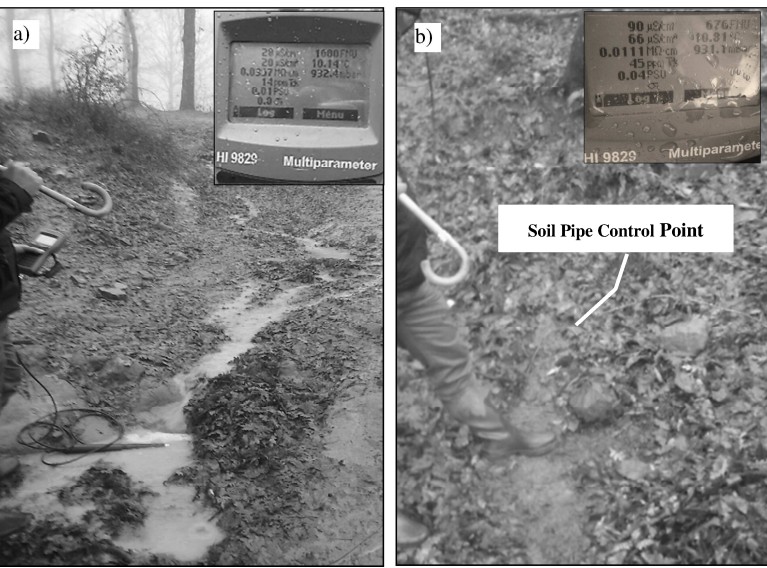

10    Figure 5. Measurements at 12:00 am in the dirty road point controls (a) and the soil pipe (b) with respective sEC measurements.

Figure 6a shows the hydro-chemograph of the storm event recorded at the main station and cumulative rainfall measured at rain gauge station near the catchment. On the plot, the phases of hydrological response in the catchment were determined by means of the progressive runoff generation activation, identified with the above-mentioned field measurements.  In Fig.6b, the

15    hysteretic Q-EC cycle (Cuomo and Guida, 2016, under revision) of the event demonstrate homogeneity in hydro-chemical





response in the rising and recession limbs. At 20.00 hrs on 29 Jan 2015, the field measurements at piezometers and Q-sEC values (approximately 60 l/s and 240 µS/cm) recorded at the main station were typical of pre-event conditions occurring during the wet period, as found by Cuomo and Guida (2016, under revision). After it started raining, in addition to the direct rainfall in main streamflow, the contribution from groundwater ridging along the riparian corridor and floodplain began to feed the

5 total discharge, the direct field evidence.. With continual rainfall, the contribution area expands and excess saturation runoff is progressively added to the discharge from colluvial hollows up to approximately Q=1000 l/s and sEC=100-120 µS/cm. In addition to these values, firstly the macropore contribution is added. Finally excess infiltration runoff from the saturated areas becomes dominant, which progressively increases the discharge, but reaches asymptotical sEC=80 µS/cm values.

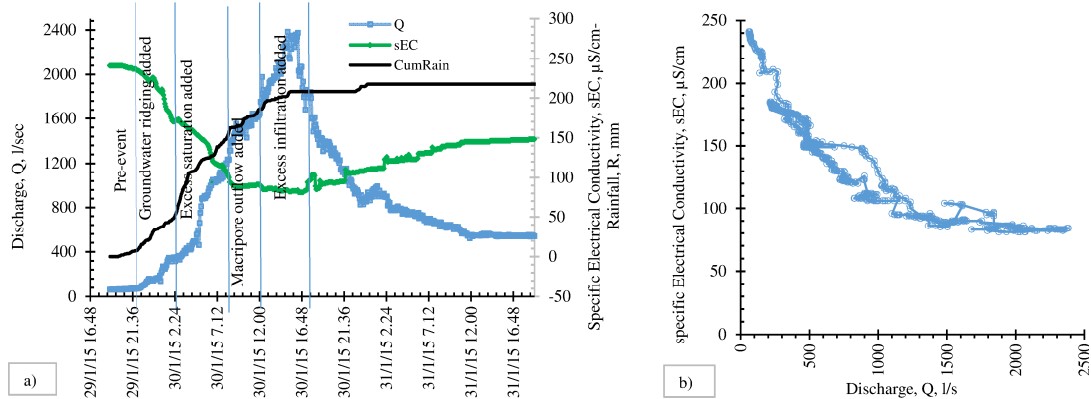

Figure 6. a. Hydro-chemograph plot of the 29-31 January 2015 storm event and related hydro-geomorphological phases, during which the runoff components are progressively added, according to Table 3; b. Q-EC hysteretic cycle of the storm event.

### 3.2 Object-based hydro-geomorphological mapping

In order to define the runoff source areas, an object-based hydro-geomorphological map of the Ciciriello catchment was created
15 using an original, automatic spatial analysis procedure. Starting from the Campania Region Technical Map (CTR), at 1:5.000 scale, a Digital Elevation Model (DEM) with a 5 mt. cell size was obtained by means of the Topo-To-Raster tool (TOPOGRID) in Arc-Gis. This algorithm provides an interpolation method specifically designed for creating hydrologically correct DEM. The grid spacing used seemed suitable for hydro-geomorphological applications since it follows the general rule that it should be adequately sufficient at the local hillslope scale, marking the transition in process dominance from hill slope to channel
20 (Peckam, 2011). Starting from this DEM, an "object-based" hydro-geomorphological map was obtained using a step-by-step rule set.

At the first step, a geomorphometric analysis was performed by combining the sine and cosine of aspect, plane and profile curvatures calculated at various cell windows: 5, 7, 9, 11, 13, 15, 17, 19 and 21 cells. The multi-scale based analysis of curvatures was performed with Landserf free GIS software, thus obtaining a raster layer for each geomorphometric calculation.
25 During the second step the best agreement with expert-based geomorphological mapping was achieved with e-Cognition software by means of an original multiresolution segmentation algorithm, by assigning a proportional increased weight to the increasing cell window size used for each raster layer (Fig. 7a). The hydro-geomorphological map (Fig. 7b) was obtained by





expert-based re-classification of the sum of plane curvature classes, choosing threshold values according to the hydrological components (hydro-geomorphotypes) listed in Table 1.

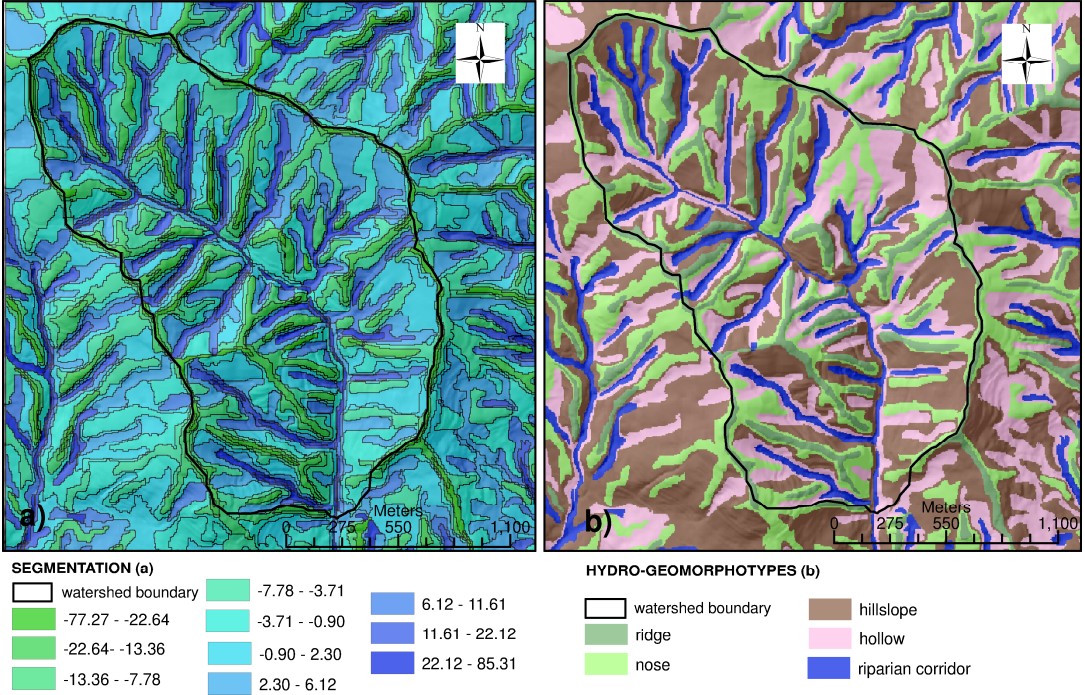

Figure 7. a) Multi-resolution segmentation map; b) Object-based hydro-geomorphological map.

Table 1: Geomorphometric classification, geomorphological correspondence, hydro-geomorphotype definition and hydro-geomorphological behaviour for each hydro-geomorphotype.

| Sum of Plan-Profile Curvature Class (PPCS) | Geomorphometric parameters and topographic position | Landform, Component or Element (Dramis et al., 2011) | Hydro-geomorphotype (HGT in Cuomo, 2012) | Hydro-geomorphological behaviour |
|---|---|---|---|---|
| SPPC < -13,4 | Convex, divergent flow-like, upslope | Upland, summit, peak, crest | Ridge | Groundwater recharge on bare bedrock and dominant excess infiltration runoff after storm |
| -13,4 >= SPPC < -3,76 | Light convex-divergent flow-like, up to midslope | Shoulder, side slope | Nose | Shallow soil, groundwater recharge area, prevalently excess infiltration runoff |
| -3,76 >= SPPC < 2,3 | Light convex-planar, parallel flow-like, midslope | Scarps, back-slope, foot-slope, wash-slope, talus, | Hillslope | Debris, deep soil, shallow aquifer, excess saturation excess and sub-surficial runoff |
| 2,3 >= SPPC < 11,6 | Planar to light concave, convergent flow-like, upslope | Glen, swallet, scar | Hollow | Deep soil, shallow aquifer, prevalently excess saturation, delayed runoff production |
| SP=C => 11,6 | Concave, convergent mid- to downslope | V-shaped stream, gully, bank, stream bed | Riparian corridor | Shallow soil, groundwater discharge, prevalently sub-surface, delayed return flow and groundwater ridging |



The classification procedure followed the criteria proposed by Hennrich et al. (1999), whose conceptual background was the 'landscape catena' (Conacher and Dalrymple, 1977), which combines surface form and pedo–hydro–geomorphological processes at hill slope scale.

In the third step, a hydrologic analysis was performed on the catchment, obtaining the contributing area map by means of the Saga module implemented in QGis. More precisely, the log-values of the contributing area map were reclassified according to the real condition of streamflow along the drainage network, observed in field during the training storm event, showed five different scenarios. Finally, a spatial statistical analysis was applied to the data from the hydro-geomorphotype map shown above and the five scenarios, in order to evaluate their spatial relationships for the training storm event that occurred on January 2015 (Fig. 6). The application at storm event time scale is described on the next section.

### 3.3 Dynamic hydro-chemograph separation

In order to understand the runoff generation that occurs during distinctive storm events for each period (wet/dry), we used the Q-EC relationship data analysis proposed by Cuomo and Guida (2013) and Guida and Cuomo (2014), considering the good accordance between the hydro-chemograph separation and the hydrograph filtering comparative procedure introduced by Longobardi et al. (2014). In particular, Cuomo and Guida (2016, under revision) subsequently proposed a modified mass balance procedure based on a "step-like", recursive, two-component hydrograph separation for the Ciciriello Catchment. The authors associated a correspondent mechanism of runoff generation to each component and the Q-EC threshold values for each mechanism in that contributing area started to enlarge and expand.

In this study, we used these values for each phase during the field survey, verifying the correspondence between the end-members hydro-chemograph signature proposed by Cuomo and Guida (2013, 2016 under revision and Guida and Cuomo, 2014) and the starting runoff contributing area.

Cuomo and Guida (2016, under revision) adopted the daily dataset illustrated in Sec. 2 (Fig. 2), using the end-members that the authors measured at the specific stormflow components by carrying out direct surveys and taking piezometric measurements. They obtained three upper and one lower boundary curves (Fig. 8), each representative of a specific mechanism, source area and timing of runoff production. The lower hyperbolic curve (LH) delimits all the Q-EC values recorded during the dry period. The upper hyperbolic (UH) curves delimit the Q-EC values that are typical of groundwater and groundwater ridging for the UH1 curves. The second upper hyperbolic curves (UH2) starts when the UH1 reaches its horizontal asymptote and the sub-surface mechanism starts. Following which the upper linear curve (UL) starts when the direct runoff and soil pipe mixes with the previous components. The estimated intersection points between the three upper consecutive curves are the Q-EC threshold values for which another mechanism starts and hydro-dynamically interacts with the previous mechanism. In this way, the waters join together before reaching the streamflow. Successively, the authors carried out the same procedure on the 13 storm events shown in Fig. 2. The events n. 1-2-3-4-10-11-12-13 were assigned to the wet recharging period while events n. 5-6-7-8-9 were assigned to the dry discharging period. Moreover, the Q-EC relationship highlights three different types of hydrologic behavior occurring in the three hydrologic periods: wet (W), dry (D) and transition (T). In this way, the boundary curves between the dry-wet and wet-transition events were obtained in order to define further inner fields. Figure 8 shows a typical "threshold hydro-geomorphological system", where each source runoff remains independent during low magnitude events, but interacts physically and functionally with other sources at higher event magnitudes, inducing superposed hydrological mechanisms and complex hydro-chemical water mixing by dilution, dispersion and diffusion. By identifying

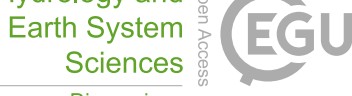



these five areas in respect to the hydrologic behavior of the catchment, it was possible to carry out the analyses for delimiting the contributing area in the next section using the thresholds listed in Table 2.

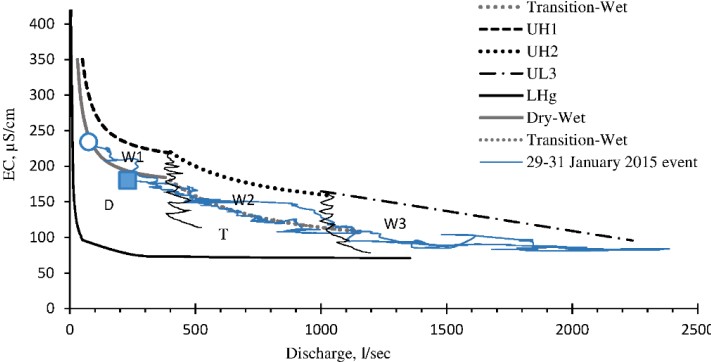

Figure 8. Delimitation of the five inner fields that define the limits of seasonal response of the catchment (modified from Cuomo and Guida, 2016, under revision) and, in blue, the hysteretic cycle of the study event, from its beginning (blue circle) to its end (blue square). Legend: UH1 and W1, upper hyperbolic curve 1 and wet area 1, respectively (typical of the Q-EC mixed value of groundwater and groundwater ridging); UH2 and W2, upper hyperbolic curve 2 and wet area 2, respectively (typical of the Q-EC mixed value of groundwater, groundwater ridging and sub-surface flow); UL3 and W3, upper linear curve and wet area 3, respectively, typical of the Q-EC mixed value of groundwater, groundwater ridging sub-surface flow and direct runoff; LH, lower hyperbolic curve typical of the Q-EC response when direct runoff is suddenly added to the groundwater, following the heavy showers occurring during the dry period; D, dry area where the Q-EC typical of a dry event falls for which only the groundwater flow feeds the streamflow; T, transition area, where the Q-EC typical values of a dry-wet or wet-dry events fall, when the groundwater flows, groundwater ridging and the soil pipe feeds the streamflow.

Table 2: Hydro-chemical parameter range, distinctive for the wet (W), dry (D) and transition (T) period events. Legend: GW is for groundwater, SSF is for subsurface flow, DR is the direct runoff. (from Guida and Cuomo, 2016 under revision).

| Field | Processes and Contributing Areas | $EC_{quick}$ Range (mS/cm) | $EC_{slow}$ Range (mS/cm) | $Q_{threshold}$ (l/s) |
|---|---|---|---|---|
| W1 | GW from bedrock deep and perched aquifer | | 250-300 | 30-50 |
| | GW+GW$_{ridging}$ added from riparian corridor | 200-220 | | 400 |
| W2 | GW+GW$_{ridging}$ along the riparian corridor | | 200-220 | |
| | GW+GW$_{ridging}$+SSF added from colluvial hollow | 120-180 | | 1000 |
| W3 | GW+GW$_{ridging}$+SSF | | 120-180 | 1000 |
| | GW+GW$_{ridging}$+SSF + DR added from soil pipe | 70-180 | | >>1000 |
| D | GW | | 320-350 | 3-5 |
| | GW+ GW$_{ridging}$ | 100-180 | | 400 |
| T | GW+ GW$_{ridging}$ | | 100-180 | 400 |
| | GW+ GW$_{ridging}$ + DR added from soil pipes | 100-120 | | |




By including the hysteretic cycle of the 29-31 January 2015 study event on the plot of Fig. 8, the hydro-geomorphological response can be classified as typical for a wet-period, that occurred after a short transition period during which the aquifer began to fill and groundwater ridging decreased progressively. As expected, during the event, all the runoff components were progressively activated when the Q-EC threshold values for each started. Consequently, the contributing areas enlarged the

floodplain upslope, the riparian corridors and the zero order basins upstream, encompassing the Q-sEC value ranges listed in Table 2. These values were verified during the field survey reported in Sec. 3.1 and used for the hydro-geomorphological analyses of the next section.

## 4     Results

For the storm study, the variability of the contribution area was obtained by combining the hydro-chemical procedure and the

object-based hydro-geomorphotype map. As a result of this analysis, contributing area space-time variability was obtained for the selected storm event by combining hydro-chemical procedure outcomes, the hydro-geomorphotype map and the contributing area scenarios.

On the right hand side of Figures 9 to 13 hydro- chemograph evolution at the five time steps discussed in Fig. 6a are illustrated, while on the left hand side of Figures 9 to 13 the progressive expanding contribution areas shown on the hydro-geomorphotype

map can be seen. Specific observations are provided in the figure captions and the corresponding values for the increasing contribution area are listed on Table 1.

Figure 9 shows pre-event conditions, when only the base flow and the decreasing groundwater ridging from previous event were activated.

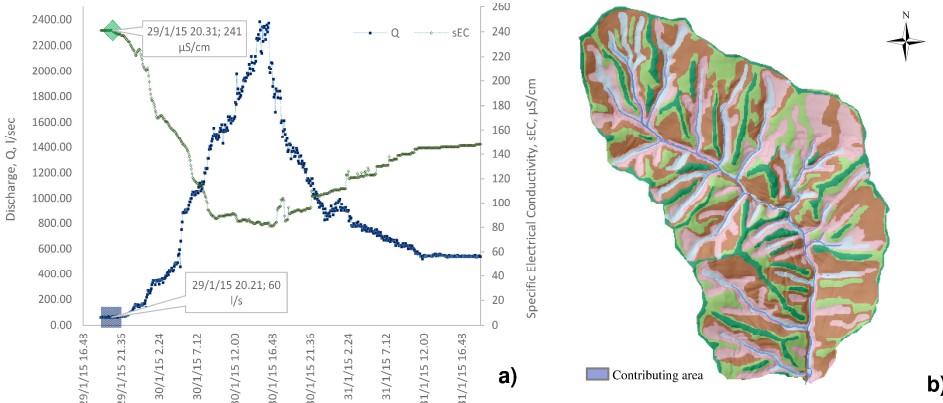

Figure 9. a) Pre-event hydro-chemograph conditions, just before the storm event, with Q=60 L/s, filled blue square and EC=240 µS/cm , filled green diamond, b) scenario corresponding to groundwater and decreasing groundwater ridging contribution to streamflow running exclusively along the riparian corridor and main streamflow.





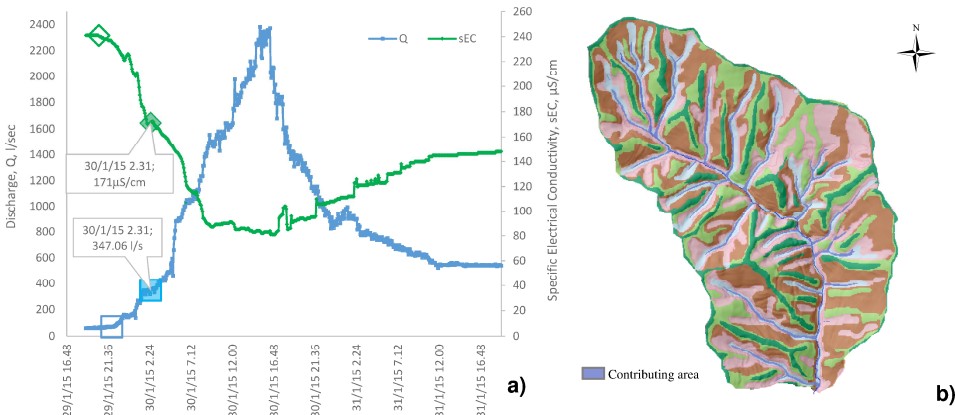

Figure 10. Initial hydro-chemograph conditions, just after the beginning of the storm event, with Q=350 L/s, filled blue square and EC=170 µS/cm, filled green diamond, b) scenario corresponding to an increasing groundwater ridging and initial saturation excess contributions to streamflow. The first occurs along the riparian corridor, the second at the apical transient channels just

5    downstream the colluvial hollows, respectively.

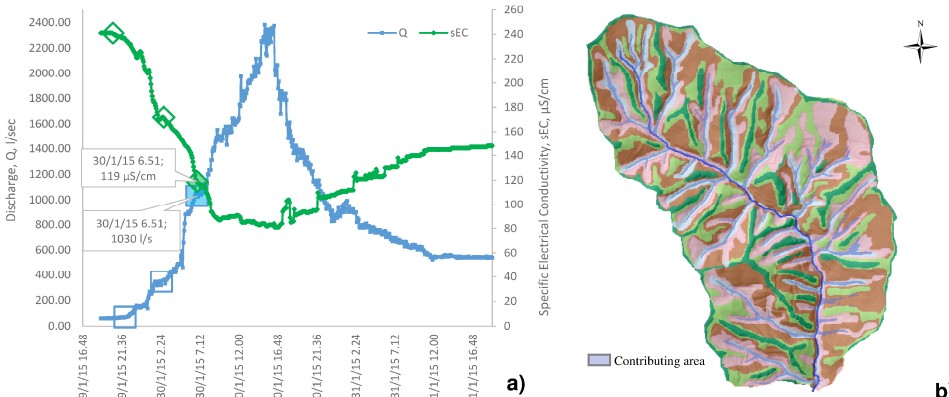

Figure 11. a) Progressive hydro-chemograph conditions, after approximately 60 mm of rainfall, with approximately Q= 1000

10    L/s, filled blue square and approximately EC=120 µS/cm, filled green diamond, b) scenario corresponding to a full saturation excess contributions to streamflow along the riparian corridor and at transient channels within the colluvial hollows, respectively.





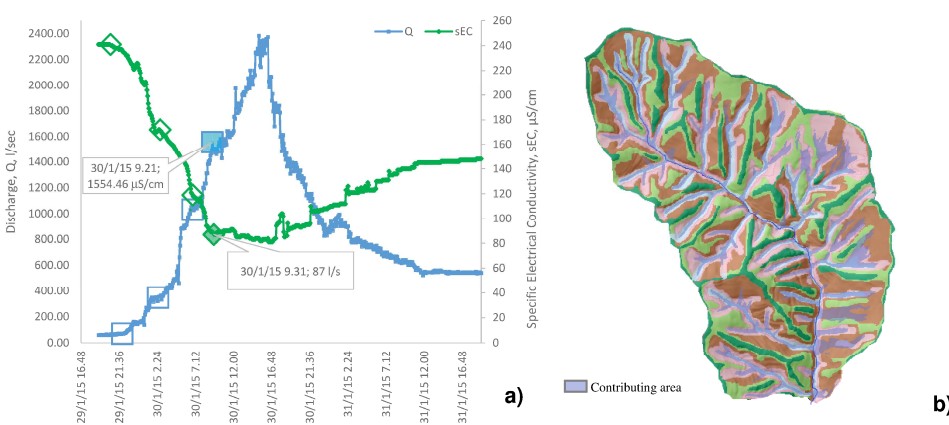

Figure 12. Advanced hydro-chemograph conditions, after approximately 80 mm of rainfall, with approximately Q= 1550 L/s,
5  filled blue square and approximately EC=90 µS/cm, filled green diamond, b) scenario corresponding to a full saturation excess
contributions to streamflow along the riparian corridor and the whole colluvial hollows, respectively.

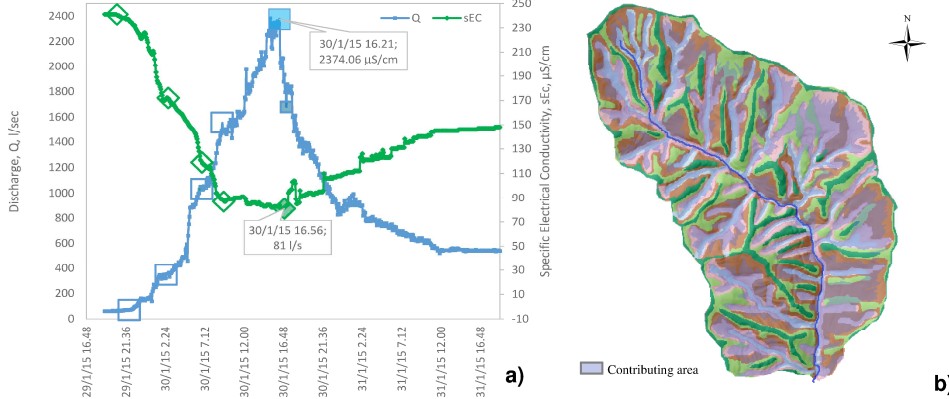

10  Figure 13. Final hydro-chemograph conditions, after approximately 100 mm of rainfall inducing a peak discharge
approximately Q= 2400 L/s, filled blue square and about EC=80 µS/cm, filled green diamond, b) corresponding both to full
saturation excess contributions to streamflow from the riparian corridor and colluvial hollows, as well as to macropore (soil
pipe and fracture) and excess infiltration, on noses and partially on the ridges respectively.





Table 3. Synoptic values of the Q-sEC scenarios and contributing areas values for each hydro-geomorphotype.

| HYDRO-GEOMORPHOTYPE (HG) | SCENARIO | DISCHARGE Q (l/s) | SPECIFIC DISCHARGE q (ls⁻¹km⁻²) | CONTRIBUTING AREA, S (m²) | % AREA 1 (A1) (on the area of each HG) | % AREA 2 (A2) (on the total area) |
|---|---|---|---|---|---|---|
| Riparian corridor | 1 | 50 | 150.93 | 56800 | 0.143 | 0.018704 |
|  | 2 | 300 | 754.67 | 102150 | 0.257 | 0.033638 |
|  | 3 | 600 | 1509.34 | 157250 | 0.396 | 0.051783 |
|  | 4 | 1000 | 2515.56 | 226775 | 0.570 | 0.074678 |
|  | 5 | 1900 | 4779.56 | 575725 | 1.448 | 0.189588 |
| Hillslope | 1 | 50 | 53.28 | 1475 | 0.00157 | 0.000486 |
|  | 2 | 300 | 319.69 | 2625 | 0.00280 | 0.000864 |
|  | 3 | 600 | 639.38 | 14525 | 0.0155 | 0.004783 |
|  | 4 | 1000 | 1065.64 | 37550 | 0.0400 | 0.012365 |
|  | 5 | 1900 | 2024.72 | 419775 | 0.447 | 0.138233 |
| Nose | 1 | 50 | 79.31 | 75 | 0.00012 | 2.47E-05 |
|  | 2 | 300 | 475.83 | 200 | 0.00032 | 6.59E-05 |
|  | 3 | 600 | 951.66 | 825 | 0.00131 | 0.000272 |
|  | 4 | 1000 | 1586.11 | 15225 | 0.0241 | 0.005014 |
|  | 5 | 1900 | 3013.60 | 118825 | 0.188 | 0.039129 |
| Hollow | 1 | 50 | 71.23 | 6975 | 0.00994 | 0.002297 |
|  | 2 | 300 | 427.37 | 15100 | 0.02151 | 0.004972 |
|  | 3 | 600 | 854.74 | 49900 | 0.07109 | 0.016432 |
|  | 4 | 1000 | 1424.56 | 93475 | 0.13316 | 0.030782 |
|  | 5 | 1900 | 2706.67 | 450075 | 0.64116 | 0.148211 |
| Ridge | 4 | 1000 | 2714.87 | 300 | 0.000814 | 9.88E-05 |
|  | 5 | 1900 | 5158.26 | 5350 | 0.0145 | 0.001762 |

5    By plotting the S vs Q data from Table 3 on a normal plot we can follow the pattern of the progressive involvement of the runoff components as specific contribution areas in streamflow (Fig. 14).

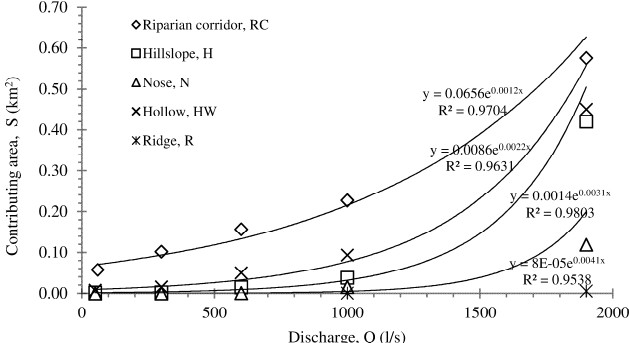

Figure 14. Plot of the Contributing Area vs Discharge from data on Table 3.





In our case, we obtained a positive exponential function for each hydro-geomorphotype curve, as shown in Fig. 14. This approach is similar to the calculations proposed by Latron (1990), but in this case the surface saturated area is calculated according to the base flow component as well as the other components connected to hydro-geomorphotypes. All the curves have a general exponential pattern (Eq. (1)):

$S(t) = S_0 e^{aQ(t)}$                    (1)

Where $S(t)$ is the total contribution area at instant t, $S_0$ the initial contribution area, $e^a$ is a constant for a specific component considered and $Q(t)$ is the discharge at time of $S(t)$.

Equation (1) can be re-written as:

$\log S(t) = aQ(t) + \log S_0$                    (2)

The riparian contribution trend is higher than the hollow and hillslope trends for a discharge from 50 to 1000 l/s, but the specific contribution areas from the latter progressively reach the same values of the riparian corridor for high discharge. In fact, a slight increase of the discharge from the riparian corridor was observed during the event (a = 0.0012). On comparing the behavior of the hollow and the hillslope, it seems that the hollow has a higher contribution area for lower discharge (from 50 to 600 l/s) than the hillslope contributing area(Fig.14). However, after the discharge increased, the two hydro-

geomorphotypes reached the same percentages as the contributing areas (% $A_2$ in Table 3) and the two exponential curves definitively intersect for a Q> 1000 l/s (Fig.14). A lower contribute originated from the nose whose contributing area is not influenced by the discharge until it reaches 1000 l/s, after which it increases rapidly (a = 0.0041).

Since 1970 authors have studied the relationships between the contributing area and the baseflow discharge (Fig.15a). In fact Ambroise (1986), Myrabo (1986) and Latron (1990) found good relationships for some catchments in which the increasing

rate of the relative saturated area decreases with the increase of a specific discharge.

Dunne et al. (1975), observed that an increase of the saturated area leads to an increase of the discharge. More recently the same relationship was observed by Martinez-Fernandez (2005). Latron and Gallarat (2007) found a linear relationship between the specific discharge and the extent of the contributing area. The authors believe that unlike the other catchments, the linear trend could be reasonable since the saturation of the catchment considered is not conditioned by its topography.

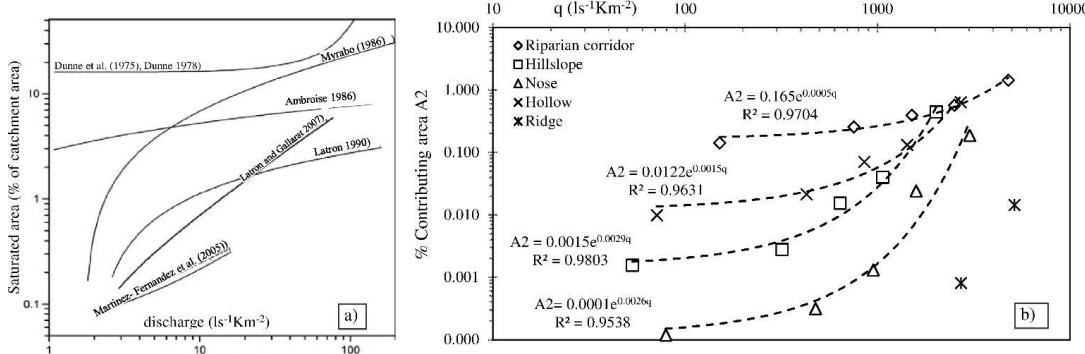

Figure 15. a) Relationship between the total extent of contributing saturated areas and the baseflow discharge in several small (less than 10 km$^2$) catchments (modified from Latron and Gallarat, 2007); b) Relationship between the contributing areas and the specific discharge for each hydro-geomorphotypes of the Ciciriello catchment.





For the Ciciriello Catchment we examined the relationships between the Percentage of the contributing area (% A2 in Table 3) and the specific discharge for each hydro-geomorphotypes considered (Fig. 15b) and we believe that this trend is similar to that observed by Dunne et al. (1975).

When a low discharge occurs, the riparian corridor starts to contribute to the increasing discharge very slowly and only for a
q = 344 l/skm$^2$ this hydro-geomorphotype widens its contributing areas. Fig. 15 shows the increase in faster contributing areas for hollow, hillslope and nose at a specific discharge q= 600, 344 and 384 l/skm$^2$, respectively. In this case these q values are considered as the q threshold values for activating runoff mechanisms.

There is an evident anomaly regarding the riparian corridor as it shows a percentage of contributing area over 100%. In our opinion, this result is due to a DEM resolution and the riparian corridor must be carefully defined due to the possible overlap
with other hydro-geomorphotypes, especially the hollows. In Fig. 15 an important result is observed concerning the intersection between all the curves at high q values. In our opinion, it is significant of the interaction between all the runoff mechanisms occurring in the catchment at high magnitude event before reaching the stream, as supposed by Cuomo and Guida (2016, under revision).

One of the more interesting results of this study is the experimental confirmation of the pre-event water contributions to stream
flow by the rapid mobilization of the capillary fringe inducing groundwater-ridging mechanisms. Despite a number of proposed processes and widespread acceptance, this mechanism is still poorly understood (Cloke et al., 2006). Therefore, this case study can be considered to be a preliminary identification, recognition and quantification at catchment scale.

## 5 Conclusion

According to the premises, the case study confirms the close link between geomorphometry and hydrology, since
geomorphometry aims to describe land surface quantitatively and  land surface is the spatial expression of the geomorphic processes acting in time and resulting in landforms generated by hydrological mechanisms, mainly in temperate and Mediterranean eco-regions. This further demonstrates how geomorphometry can usefully support hydrological analysis, by improving an interdisciplinary potential for future developments in connecting hydrology and geomorphology in data acquisition, mapping, analysis modeling and general purpose applications. This is the purpose of object-based hydro-
geomorphology, based on the methods for recognizing and classifying distinctive hydro-objects within catchments, attaching ontology and semantics to significant catchment areas with distinctive hydrological behavior and response in order to allow for their objective description, holistic analysis and inter-catchment comparison.

In this perspective, firstly by means of a recursive training-target approach (Guida et al., 2015) we verified a good agreement between the expert-based geomorphological mapping and the object-based geomorphometric map, obtained by a weighted
profile and plane curvature sum.,.

Therefore, by combining the hydro-chemical analysis and the object-based hydro-geomorphotype map, the variability of the contribution area during a significant storm event was spatially modeled using the log-values of the flow accumulation. In spite of its simplicity, this parameter provided better statistical fit with the observed contribution areas detected during the event by means of direct surveys and discharge/groundwater measurements. The runoff components were determined for that
storm event and specific runoff discharge from each contributing hydro-geomorphotype was calculated for each time step on the hydro-chemograph.



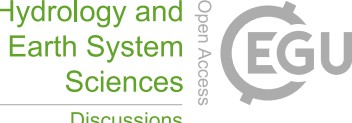

This study is the experimental confirmation of the role and entity of pre-event water contributions to stream flow by the rapid mobilization of the capillary fringe inducing the groundwater ridging mechanism in step sloping terrains. Despite a number of proposed processes and widespread acceptance, this mechanism is still poorly understood (Cloke et al., 2006) therefore this case study can be considered as a preliminary identification, recognition and quantification of this particular mechanism at

catchment scale,. According to Marcus et al. (2004), this study emphasizes the fact that field-based process studies must "*continue to form the underpinning of hydrologic application in GIS's*" and "*GIScience should not come at the expense of sacrificing field-based studies of hydrologic processes and responses*".

This is an approach that can fill the gap between simple lumped hydrological models and sophisticated hydrological distributed models based on numerous quantitative parameters and expensive data collection. This kind of interdisciplinary and integrate

approach can be usefully applied to similar, rainfall-dominated, forested and no-karst catchments in the Mediterranean eco-region by using a inexpensive, parsimonious and effective methodology, as suggested by the Biosphere2 Program for water resource assessment and management. In fact, in UNESCO International Designation Areas (such as the Cilento Global Geoparc), hydro-geodiversity must be guaranteed by the Global Geopark Network Mission according to the requirements laid down by the World Heritage Cultural Landscape Management and natural and managed ecosystems (A1) must be safeguarded

as established by the MANANDBIOSPHERE Program.

In this perspective, geomorphometry plays a fundamental role in quantifying and objectively mapping hydro-geomorfological entities with hydrological relevance that require monitoring and modeling in production, transfer and routing flow between the different units in the catchments, as the knowledge base for progressive ecological planning on the sustainable use of water resources and best practices in land use improvements.

**Acknowledgments**

The paper was financed with ORSA155417, University of Salerno research funds. The authors would like to thank Lovisi Pasqualino for field measurements, Benevento Giuseppe for scientific support (CUGRI), Aloia Aniello and Angelo De Vita, Cilento Global Geopark manager and director, for their financial support and Biafore for the rainfall data obtained from the Campania region Monitoring System.

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
