# Peer review of "Using object-based geomorphometry for hydro-geomorphological analysis in a Mediterranean research catchment"

_Hydrology and Earth System Sciences, 2016_

## Referee Comment (RC1) · Anonymous Referee #1 · 16 Mar 2016

The paper presents interesting results of application of the object-based geomorphometry for hydrological analysis and therefore is within a scope of the HESS journal and worth of publication, especially in this special issue. Further, the study impressively combines detailed field work aimed at monitoring and collecting of several hydrological data within the research catchment (complemented with authentic pictures) with modern methods of digital terrain analysis, showcasing how useful linking these two disciplines can be.

All the results show sufficient support to the interpretations and conclusions. Especially the results of the modelling of variability of the contribution area based on combining field data from a selected storm event with the "object-based hydro-geomorphotype

map" are very interesting. Most of the experiments and methods are described precisely, except for the part about segmentation resulting in "hydro-geomorphotypes" objects in the classification process (detailed comment please see in the Specific comments and suggestions section).

The authors give proper credit to the related hydrological work with clear indication of their new contribution, but some theoretical background about object-based image analysis and multi-resolution segmentation as well as some previous work related to the application of this method in geomorphology or geomorphometry should be mentioned (e.g. important papers which encouraged authors to apply it). Suggested literature is mentioned in the Potentially useful literature section. The title clearly reflects the contents of the paper, however, I might suggest for consideration adding the term "object-based" before "geomorphometry" for more precise specification.

Apart from the small errors mentioned in the Technical corrections section, the paper is well written and structured with reasonable figures and tables. Overall, it is written in good and understandable English, although it could benefit from reading and corrections by a native English speaker.

Specific comments and suggestions:

1. Fig. 1 is a bit difficult to read and understand, especially some features of the "Monitoring system". Adding colours at least to these symbols would improve the readability.

2. Although, according to the authors and the stated reference (Peckham, 2009), use of grid spacing of 5 meters seems suitable (page 7, line 18), in the Results section the authors state that contribution area anomaly of the riparian corridor could be attributed also to the DEM resolution (page 16, line 8). There are other methods how to correctly determine DEM resolution (e.g. Hengl, 2006). Resolution of 2.5 meter could be calculated using the simplest equation in Hengl (2006) based on the working scale. Of course, increased resolution would increase computation time of other analyses (especially segmentation), but this anomaly might be avoided this way. I would suggest to

at least mention it in the discussion.

3. Page 7, line 24 as well as page 16, line 29: Was the mentioned expert-based geomorphological map created based on the segmentation input layers or taken from a previous study? Either way, it would be useful to have it described there and perhaps even more useful to display it as a figure to directly see the mentioned agreement between these two maps.

4. Several crucial pieces of information about the performed multi-resolution segmentation (page 7, starting in line 25) are missing, especially the value of scale parameter and a method of its determination. For readers would be also useful to know the used values of shape and compactness parameters. Were the aspect layers used as an input into the segmentation or only the plan and profile curvature, whose values were later used in the classification? It is not clear from the text.

5. Are the objects resulting from segmentation displayed in Fig. 7 (A) classified based on a sum of plan and profile curvature? If yes, I would suggest to mention it in the figure caption.

6. I would suggest to use darker tone of colour for "contributing area" or perhaps to add black outline to make it more readable in Fig. 9 - 13.

Technical corrections:

- page 1, line 20: I would replace "plane" with "plan" to have the correct term for this curvature. Please apply also in the rest of the text

- page 6, line 3: I would replace abbreviation "zob" with full "zero order basin" as it is in the figure under it or in page 3, line 14 or page 11, line 5

- page 7, line 5: typing error "." at the end of the first sentence should be removed

- page 7, line 16: I suggest to replace "5 mt. cell size" with "5-meter cell size"

- page 7, line 17: I suggest to replace "Arc-Gis" with "ArcGIS" as it is the official name

- page 7, line 20: there is a reference to (Peckam, 2011) but in the list of references is (Peckam, 2009), please correct it; I would replace "Starting from this DEM" with e.g. "This DEM was used in"

- page 7, line 25: I suggest to replace "e-Cognition" with "eCognition Developer" as it is the official name

- page 8, line 4: caption for Fig.7, please be consistent with the name of the segmentation. Here is "multi-resolution", in the previous text is "multiresolution"

- page 8, Table 1: I think that in the last row it should be "SPPC" instead of "SP=C"

- page 9, line 5: I suggest to replace "Saga" with "SAGA" and "QGis" with "QGIS" as these are the official names

- page 10, Fig. 8: I would say there is one extra "Transition-Wet" in the legend, otherwise it is not recognisable in the figure if it should represent other type of transition area

- page 11, line 16: I think there should be "Table 3" instead of "Table 1"

- page 12, line 1 and 2: caption of Fig. 10, there should be added "approximately" before Q and EC values as it is in captions of Fig. 11 – 13

- page 16, line 30: typing error "," at the end of sentence should be removed

Potentially useful literature:

Anders, N.S., Seijmonsbergen, A.C., Bouten, W., 2011. Segmentation optimization and stratified object-based analysis for semi-automated geomorphological mapping. Remote Sens Environ 115, 2976–2985. doi:10.1016/j.rse.2011.05.007

Baatz, M., Schäpe, A., 2000. Multiresolution Segmentation : an optimization approach for high quality multi-scale image segmentation, in: Strobl, J., Blaschke, T., Griesebner, G., Wichmann-Verlag, H. (Eds.), Angewandte Geographische Informationsverarbeitung XII. pp. 12–23.

Dragut, L., Blaschke, T., 2006. Automated classification of landform elements using object-based image analysis. Geomorphology 81, 330–344. doi:10.1016/j.geomorph.2006.04.013

Dragut, L., Csillik, O., Eisank, C., Tiede, D., 2014. Automated parameterisation for multi-scale image segmentation on multiple layers. ISPRS J Photogramm 88, 119–127. doi:http://dx.doi.org/10.1016/j.isprsjprs.2013.11.018

Dragut, L., Minár, J., Csillik, O., Evans, I.S., 2013. Land-surface segmentation to delineate elementary forms from Digital Elevation Models, in: Geomophometry 2013. pp. 2–5.

Eisank, C., Smith, M., Hillier, J., 2014. Assessment of multiresolution segmentation for delimiting drumlins in digital elevation models. Geomorphology 214, 452–464. doi:10.1016/j.geomorph.2014.02.028

Hengl, T., 2006. Finding the right pixel size. Computers & Geosciences 32, 1283–1298. doi:10.1016/j.cageo.2005.11.008

van Asselen, S., Seijmonsbergen, A.C., 2006. Expert-driven semi-automated geomorphological mapping for a mountainous area using a laser DTM. Geomorphology 78, 309–320. doi:10.1016/j.geomorph.2006.01.037

---

## Author Comment (AC1) · 23 Mar 2016

Thank you for yours considerations about our article. In order to benefit of yours suggestions, we are revising the paper following yours specific comments and technical corrections. Best regards Domenico Guida and co-authors

---

## Referee Comment (RC2) · Anonymous Referee #2 · 7 Apr 2016

Generally well written paper fully within the scope of HESS. It presents modern hydrogeomorphological approach investigating space-time variability of runoff process including innovative combination of hydrochemograph analyses and geomorphometric land surface classification. Because the original hydrochemograph analyses should be independently published (Cuomo, Guida, under revision) it is possible to suppose that geomorphometric classification and interpretation of results are main innovative moment. Unfortunately the process of used land surface segmentation is not sufficiently explained and justified. Using of sum of weighted plan and profile curvature is declared in the Abstract and Conclusion, but only weighted plan curvature is mentioned on p. 8 and no specification of the procedure is done. Production of flow accumulation maps is

unclear too as well as specifications of the multisegmentation algorithm and criterion of accordance between expert based mapping and multiresolution segmentation. Table 3 and Fig. 14 presents a synthesis of results obtained. However majority of variables are not explained in the table. Systematic shift of extreme values in Fig 14 could point to systematic underestimation of contributing area in 5th scenario, but it is not discussed as well as quality of geomorphometric procedures and results in general. . Particulars, some other vagueness, mistakes and other problems are in attached commented version of pdf. Without clarification of marked problems is not possible fully evaluate the paper. However the title I feel as appropriate, abstract, structure, supplementary material and language are generally OK. Formal insufficiency are marked in the pdf.

Please also note the supplement to this comment:
http://www.hydrol-earth-syst-sci-discuss.net/hess-2016-68/hess-2016-68-RC2-supplement.pdf

**Supplement:**

[revised manuscript text omitted]

---

## Author Comment (AC2) · 20 Apr 2016

Dear referee, First of all, thank you for your useful suggestions!

Following your statement: "Without clarification of marked problems is not possible fully evaluate the paper", we wrote this reply in order to give you some explication which could be useful to continue the revision process:

1. "The process of used land surface segmentation is not sufficiently explained and justified. . . . . . . . . . ..as well as specifications of the multisegmentation alghoritm. . . ."

- We kindly explain that we used different parameters for the segmentation and for the classification, but may be we didn't well explained in the text and in the figure 3. - For

multiresolution segmentation algorithm it was used: A) the sine and cosine of aspect and B) the "weighted" plan and profile curvature.

- For classification was used only plan curvature (as it is mentioned on p. 7 lines 27-28).

- Parameters used for multiresolution segmentation are: scale 7, shape 0.0002, compactness 0.0002.

2. "Production of flow accumulation maps is unclear too"

- The flow accumulation map was obtained using "Catchment area" algorithm available in the SAGA module implemented in QGIS; - in the text we used alternatively "flow accumulation" and "contributing area" map (see pag. 5 lines 10-11).

3. Criterion of accordance between expert based mapping and multiresolution segmentation

- With the term "Expert based geomorphological map" we mean a traditional "geomorphological map" performed by geomorphologist, who drawn polygonal features on a topographic map (CTR 1:5.000 vector data map). - Thus, the criterion of accordance between the geomorphological map and multiresolution segmentation is the training-target procedure proposed in the section "Methodology" of Guida et al.2015 (see reference).

4. "Majority variable are not explained in the table 3".

- the specific discharge "q" is calculated from the area of each hydro-geomorphotype, that will be added into the final revised text

- The AREA 1 and AREA 2 haven't any relation with the areas of fig.8

- The AREA 1 and AREA 2 are used for comparison with others studies and to explain the saturation state both for each hydro-geomorphotype and the catchment

- AREA 1: is the ratio (it is not in %) between the contributing area and the area of each

hydro- geomorphotype. This ratio is used in the fig. 15b pag. 15 (there is an error in the figure 15 b the area is A1 and not A2)

- AREA 2: is the ratio (it is not in %) between the contributing area and the area of the basin. This ratio is used in the text pag. 15 line 15

5. "Systematic shift of extreme value in fig 14 could point to systematic underestimation of contributing area in the 5th scenario".

- We tried different curves according to Fig. 15 a in order to compare our results to the others cited studies. - In particular, the curve adopted is comparable with that of Dunne et al., 1975 and Dunne, 1978.

6. Quality of geomorphometric procedure: - The quality of the geomorphometric procedure will be improve using a DEM with a higher resolution than that used in this study.

Answers to some marked questions in the pdf:

Pag. 5 line 9: Please, define / quote this algorithm

- The statement "hydrologically-corrected DEM was obtained by means of the D∞ algorithm" is not corrected - We must replace the above sentence in " hydrologically-corrected DEM was obtained by means of the "Fill sink" tool of ArcHydro geospatial data model designed to operate within ESRI's ArcInfo software.

Pag. 7 lines 22-23: Please explain choice of this variables. Why you consider aspect as more important as e.g. slope gradient?

- We didn't consider slope gradient because, a part for the valley bottom and hilltop, it is quite constant and didn't give additional information to the segmentation procedure.

Pag. 7 lines 23-24: Why only curvatures were used in following analysis? Why not aspect?

- The aspect was not processed in the Landserf free Gis because we just obtained a good accordance between the training and the target segments by using the sine and cosine aspect parameters in addition to the curvatures analyses.

Pag. 7, line 25: Is it Expert-based contributing area map from Fig 3? Please, be terminologically consistent

- in the text we used alternatively "expert based geomorphological map" and "expert based contributing area map".

Pag. 7, line 25-26-27: How was the multisegmentation algorithm used, what variants were tested and what was a criterion of agreement of expert based mapping and multiresolution segmentation?

- We have tested the algorithm trying different incremental weight (1, 2, and so on) for the plane and profile curvatures (i.e for incremental weight of 1 of plane/profile curvature with cell window 5 – weight 1; cell window 7- weight 2 …. Cell window 21- weight 10). - In the final revision procedure we will insert a table with the weight

Pag. 8 Line 5-6-7: Not clear. Please, describe in more detail

- In order to obtain the five different scenarios showed in fig 9-13, the log of contributing area map (flow accumulation map) has been reclassified according to the real condition of water flow observed in field.

Pag. 8 line 28: How was the sum of plane curvature classes computed? Why only plane and not profile curvature was used - We used E-Cognition to sum, for each segment derived from the segmentation, the plane curvature computed with different windows. - Was used only the plane curvature in the classification because was enough to classify the hydro-geomorphotype

Pag.14 line 9: Used exponential relation evidently systematically overestimate extreme values. Because extremely low number of values high R2 cannot be substantive. Were tested another types of relationships? If the exponential relationships result from the

hydrological theory assesment of contributing area could be questionable.

- We tried different curves according to Fig. 15 a in order to compare our results to the others cited studies. - The exponential law was used because described better than the linear and others curves the five scenarios.

Pag. 16 line 29: No info about weighted profile and plan curvature sum computation. Moreover on p. 8 only plan curvature is mentioned as used for segmentation!!

- The corrected statement is: "a weighted plane curvature sum".

Pag. 16 Line: Where is it documented?

- See Pag. 8 Line 5-6-7 - We used the logarithm of flow accumulation to scale it to a more condensed and linear range. An example is the "Topographic Index" (Quinn et al., 1991, 1995).

Pag. 16 line 33: Not documented

- The sentence "this parameter provided better statistical fit with the observed contribution areas detected during the event by means of direct surveys and discharge/groundwater measurements" is modify in " spatial distribution of this parameters offered a good accordance with the observed contribution areas detected during the event by means of direct surveys and ischarge/groundwater measurements"

We hope that the above clarifications are useful to continue the revision process. Write us for any others informations you need!

Best regards

The Authors

---

## Author Comment (AC3) · 23 Jun 2016

I would inform you that the paper under revision cited in the submitted article, is now accepted.

I attached to the present the proof version.

Best regards

Albina Cuomo

[Figure]
[Figure]

**Using hydro-chemograph analyses to reveal runoff generation processes in a Mediterranean catchment**

**A. Cuomo[1] and D. Guida[1]**

[1]{Department of Civil Engineering, University of Salerno, Via Giovanni Paolo II, 132, 84084, Fisciano (SA) Italy}

Correspondence to: A. Cuomo (acuomo@unisa.it, 0039 089964119)

**Fig. 1.**

[Figure]

---

## Author Response (AR1)

RESPONSES TO EDITOR AND REVIEWERS COMMENTS (colored text)

I have examined the manuscript as well as the reviewers' comments and authors' reply of these. I do think that a set of revisions is advised. During this round of reviewers is it not my intention to discount any of the comments raised by Reviewers.

> Authors warmly appreciate and thank the editor.

**Referee n. 1**

-The authors give proper credit to the related hydrological work with clear indication of their new contribution, but some theoretical background about object-based image analysis and multi-resolution segmentation as well as some previous work related to the application of this method in geomorphology or geomorphometry should be mentioned  (e.g. important papers which encouraged authors to apply it). Suggested literature is mentioned in the Potentially useful literature section".

> We introduced in the chapter "Introduction" (page 2 lines 15-17, marked-up manuscript) the theoretical background that you have suggested.

The title clearly reflects the contents of the paper, however, I might suggest for consideration adding the term "object-based" before "geomorphometry" for more precise specification.
> We added to the title the term "object-based" before "geomorphometry"

*Specific comments:*

1. Fig. 1 is a bit difficult to read and understand, especially some features of the "Monitoring system". Adding colours at least to these symbols would improve the readability.

> We improved the readability of the features in the map, by changing the colours and the shapes of the monitoring stations (Fig. 1 new)

2. Although, according to the authors and the stated reference (Peckham, 2009), use of grid spacing of 5 meters seems suitable (page 7, line 18), in the Results section the authors state that contribution area anomaly of the riparian corridor could be attributed also to the DEM resolution (page 16, line 8). There are other methods how to correctly determi ne DEM resolution (e.g. Hengl, 2006). Resolution of 2.5 meter could be calculated using the simplest equation in Hengl (2006) based on the working scale. Of course, increased resolution would increase computation time of other analyses (especially segmentation), but this anomaly might be avoided this way. I would suggest to at least mention it in the discussion.

> We added this concept on page 11 line 17-18 and page 12 line 1 (in the marked-up manuscript)
> In the scientific literature some methods are known for a more suitable grid resolution (Hengl, 2006) based on the properties of the input data (i.e complexity of the terrain), but the grid spacing used seemed suitable for hydro-geomorphological applications since it follows the general rule that it should be adequately sufficient at the local hillslope scale, marking the transition in process dominance from hill slope to channel (Peckham, 2009)."

3. Page 7, line 24 as well as page 16, line 29: Was the mentioned expert-based geomorphological map created based on the segmentation input layers or taken from a previous study? Either way, it would be useful to have it described there and perhaps even more useful to display it as a figure to directly see the mentioned agreement between these two maps.

The expert-based geomorphological map was created by direct field surveys of an expert geomorphologist. We displayed the expert-based geomorphological map in the fig. 7 new (ex fig. 7) where in the previous version only the object-based morphological map was shown.

4. Several crucial pieces of information about the performed multi-resolution segmentation (page 7, starting in line 25) are missing, especially the value of scale parameter and a method of its determination. For readers would be also useful to know the used values of shape and compactness parameters. Were the spect layers used as an input into the segmentation or only the plan and profile curvature, whose values were later used in the classification? It is not clear from the text.

We added all this information on page 12 from line 10 in the marked-up manuscript and we added a new table 1

5. Are the objects resulting from segmentation displayed in Fig. 7 (A) classified based on a sum of plan and profile curvature? If yes, I would suggest to mention it in the figure caption.

For classification was used only plan curvature see Fig. 7a

6. I would suggest to use darker tone of colour for "contributing area" or perhaps to add black outline to make it more readable in Fig. 9 - 13.

We used darker tone of colour for "contributing area"

Technical corrections:

- page 1, line 20: I would replace "plane" with "plan" to have the correct term for this curvature. Please apply also in the rest of the text
    We replaced "plane" with "plan"

- page 6, line 3: I would replace abbreviation "zob" with full "zero order basin" as it is in the figure under it or in page 3, line 14 or page 11, line 5

    We replaced the abbreviation "zob" with "zero order basin"

- page 7, line 5: typing error "." at the end of the first sentence should be removed

    Removed

- page 7, line 16: I suggest to replace "5 mt. cell size" with "5-meter cell size"
    Done

- page 7, line 17: I suggest to replace "Arc-Gis" with "ArcGIS" as it is the official name
    Done
- page 7, line 20: there is a reference to (Peckam, 2011) but in the list of references is (Peckam, 2009), please correct it; I would replace "Starting from this DEM" with e.g. "This DEM was used in"

    Done

- page 7, line 25: I suggest to replace "e-Cognition" with "eCognition Developer" as it is the official name

> Done

- page 8, line 4: caption for Fig.7, please be consistent with the name of the segmentation. Here is "multi-resolution", in the previous text is "multiresolution"

> Done

- page 8, Table 1: I think that in the last row it should be "SPPC" instead of "SP=C"

> Done

- page 9, line 5: I suggest to replace "Saga" with "SAGA" and "QGis" with "QGIS" as these are the official names

> Done

- page 10, Fig. 8: I would say there is one extra "Transition-Wet" in the legend, otherwise it is not recognisable in the figure if it should represent other type of transition area

> There was an error in the figure 8. We cancelled the second "Transition-wet" legend (fig. 8 new)

- page 11, line 16: I think there should be "Table 3" instead of "Table 1"

> In the new version of the paper we added a new table, for this reason we changed in "Table 4" (there is a new renumbering).

- page 12, line 1 and 2: caption of Fig. 10, there should be added "approximately" before Q and EC values as it is in captions of Fig. 11 – 13

> Added

- page 16, line 30: typing error ",." at the end of sentence should be removed

> Removed

**Referee n. 2**

1. "The process of used land surface segmentation is not sufficiently explained and justified............as well as specifications of the multisegmentation alghoritm...."

   - We optimized this part of the text see page 10 from line10 and added a new Table 1 (marked-up manuscript)
   -

2. "Production of flow accumulation maps is unclear too"

   The flow accumulation map was obtained using "Catchment area" algorithm available in the SAGA module implemented in QGIS;

3. Criterion of accordance between expert based mapping and multiresolution segmentation

   With the term "Expert based geomorphological map" we mean a traditional "geomorphological map" performed by geomorphologist, who drawn polygonal features on a topographic map (CTR 1:5.000 vector data map).
   Thus, the criterion of accordance between the geomorphological map and multiresolution segmentation is the training-target procedure proposed in the section "Methodology" of Guida et al.2015 (see reference).

4. "Majority variable are not explained in the table 3".

   The Table 3 is now renumbering in Table 4. We explained the variables in the caption of the table.

5. "Systematic shift of extreme value in fig 14 could point to systematic underestimation of contributing area in the 5$^{th}$ scenario".

   We tried different curves according to Fig. 15 a in order to compare our results to the others cited studies.
   In particular, the curve adopted is comparable with that of Dunne et al., 1975 and Dunne, 1978.

6. Quality of geomorphometric procedure:
   The quality of the geomorphometric procedure will be improve using a DEM with a higher resolution than that used in this study.

Answers to some marked questions in the pdf:

Page 3 Fig. 1: It is problematic to distinguis colluvial and alluvial soils in the map. No Main station and only one Sub station is in the map following legend (control size of symbols in the map and in the legend, please).

   We improved the readability of the features in the map, by changing the colours and the shapes of the monitoring stations (Fig. 1 new).

Page 4 line 10: Grey columes are not explained (wet periods?) as well as full black and gray lines (Q and EC?). Scale on the left y axis is (I suppose) for log Q. Where is scale for Specific Electrical Conductivity?

   Grey columes are introduced only for a clear visualization of the events
   The significance of the full black and gray lines correspondent to Q and EC respectively and are shown in the legend that is inside the fig. 2

The scale on the left y axis is for Log Q as indicated in the figure 2
The scale for the Specific Electrical Conductivity is the same of the Discharge.

Page 5 line 6: sEC? - abbrev EC was only established above.

we added statement "(we used either sEC or EC in the following)" on page 5 lines 6 in the marked-up manuscript

Page 5 line 9: Please, define / quote this algorithm

The statement "hydrologically-corrected DEM was obtained by means of the D∞ algorithm" is not corrected
We replaced the above sentence in " hydrologically-corrected DEM was obtained by means of the "Fill" tool of ESRI's ArcInfo software.

Page 5 line 10: they are not mentioned in Fig. 3 and anywhere above - add it ?into second middle block: "DEM ( 5x5) hydrological D∞corrected, flow accumulation maps

We modify the flow chart and added the "flow accumulation map" in the second middle block.

Page 6 line 3: What is "zob springs"? Zero Order Basin springs - after Fig. 4? Do not use abbrev uselessly.

Changed the "Zob spring" in "Zero order basin spring"

Page 6 line 4: Dot

Done

Page 6 Fig. 4: Complete captions for second photo, distinguish a) and b).

We completed the caption

Page 6 Fig. 5: Only abbrev EC was introduced

Corrected in the caption

Page 6 line 15: It is not fully correct to quote not published paper introducing a new concept.

The paper is now accepted for publication

Page 7 line 5: ?? formulation

Cancelled

Page 7 line 11: a)

Done

Page 7 line 12 : b); Change of colour of the line could show rising and recession limbs of the cycle.

Done

We changed the colour of the line (Fig. 6 new)

Page 7 lines 15-17: Does it means by digitizing of contourlines ot this map?

The CTR (regional thecnical map) 1:5000 is a CTR 1:5.000 is a vector data (CAD format) topographic map with contour lines provided of z coordinates (elevation values).

Page 7 lines 22-23: Please explain choice of this variables. Why you consider aspect as more important as e.g. slope gradient?

We didn't consider slope gradient because, a part for the valley bottom and hilltop, it is quite constant and didn't give additional information to the segmentation procedure.

Page 7 lines 23-24: Why only curvatures were used in following analysis? Why not aspect?

The aspect was not processed in the Landserf free Gis because we just obtained a good accordance between the training and the target segments by using the sine and cosine aspect parameters in addition to the curvatures analyses.

Page 7, line 25: Is it Expert-based contributing area map from Fig 3? Please, be terminologically consistent

in the text there is an error. In order to be terminologically consistent we revised the flow chart and the correct term is "expert based geomorphological map"

Page 7, line 25-26-27: How was the multisegmentation algorithm used, what variants were tested and what was a criterion of agreement of expert based mapping and multiresolution segmentation?

We have tested the algorithm trying different incremental weight (1, 2, and so on) for the plan and profile curvatures (i.e for incremental weight of 1 of plane/profile curvature with cell window 5 – weight 1; cell window 7- weight 2 …. Cell window 21- weight 10).
We added a new table where are listed all the weight (Tab. 1 new)
the criterion of accordance between the geomorphological map and multiresolution segmentation is the training-target procedure proposed in the section "Methodology" of Guida et al.2015 (see reference).

Page 8 line 28: How was the sum of plane curvature classes computed? Why only plane and not profile curvature was used
We used E-Cognition to sum, for each segment derived from the segmentation, the plane curvature computed with different windows.
Was used only the plane curvature in the classification because was enough to classify the hydro-geomorphotype

Page 9 lines 5-7: Not clear. Please, describe in more detail.

We changed the statement in: More precisely, the log-values of the  flow accumulation map was reclassified according to the real conditions observed in streamflow and in each hydro-geomorphotypes during five different scenarios of the training storm event.

Page 9 line 24: ?LHg
Corrected
Page 10 line 10: LHg in the Figure?

We changed "LH" in "LHg"

Page 10 line 11: Dry-Wet in the Figure?

Dry-Wet in the Figure is the shared line between the Dry and the Wet areas. In the text we means the Dry area

Page 10 line 12: Transition-Wet in the Figure? There are two symbols for it! (first and last but one)

Transition-Wet in the Figure is the shared line between the Wet and the Tansition areas. In the text we means the transition area

Page 12 figures 10: It could be usefull mark it in the map - Contributing area seems to be same as in Fig. 9.

We used darker tone of colour for "contributing area" and orange colour for the riparian corridor.

Page 14 Table 3: From what values of area was computed? (Evidently no from Contributing area)?

q is the specific discharge calculated respect the area of the catchment

Use km$^2$ to be comparable with specific discharge.

Done

What is AREA 1 and AREA 2? Is there any relation with wet area 1 and wet area 2 in Fig. 8 (also not explained)? How they are computed? Are they really %? (very small values)

The AREA 1 and AREA 2 haven't any relation with the areas of fig.8

The AREA 1 and AREA 2 are used for comparison with others studies and to explain the saturation state both for each hydro-geomorphotype and the catchment

AREA 1: is the ratio (it is not in %) between the contributing area and the area of each hydro- geomorphotype.
This ratio is used in the fig. 15b (there is an error in the figure 15 b the area is A1 and not A2)

AREA 2: is the ratio (it is not in %) between the contributing area and the area of the basin.
This ratio is used in the text page 24 line 15.

Pag24 line 9: Used exponential relation evidently systematically overestimate extreme values. Because extremely low number of values high R2 cannot be substantive. Were tested another types of relationships? If the exponential relationships result from the hydrological theory assesment of contributing area could be questionable.

We tried different curves according to Fig. 15 a in order to compare our results to the others cited studies.
The exponential law was used because described better than the linear and others curves the five scenarios.

Page 15 line 2: French written publication - the principle should be briefly repeated.

The correct reference is Latron and Gallart (2007)

Page 15 Line 15: No intersection is on the figure unless extrapolation the curves behind measured values.

removed

Page 16 line 29: No info about weighted profile and plan curvature sum computation. Moreover on p. 8 only plan curvature is mentioned as used for segmentation!!

We cancelled "weighted profile and plan curvature sum computation"
We rewrote this part of the text see page 12 line 6 of the marked-up manuscript. Please see the new table 1

Page 16 Line : Where is it documented?

We used the logarithm of flow accumulation to scale it to a more condensed and linear range. An example is the "Topographic Index" (Quinn et al., 1991, 1995).

Page 16 line 33: Not documented

[revised manuscript text omitted]
 was reclassified according to the real conditions observed in streamflow and each hydro-geomorphotypes during five different scenarios occurred during the training storm event. The best accordance between the reclassified log-values of the flow accumulation map and the field evidences represents the final Contributing Area scenarios map.

**3.2 Object-based hydro-geomorphological mapping**

In order to quantitatively define the runoff source areas, an object-based hydro-geomorphological map of the Ciciriello catchment was created using an original, automatic spatial analysis procedure. Starting from the Campania Region Technical Map at 1:5.000 scale (CTR), a vector map provided of elevation values), at 1:5.000 scale, a Digital Elevation Model (DEM) with a 5--mteter- cell size was obtained by means of the Topo-To-Raster tool (TOPOGRID) in Arc-GisGIS. This algorithm provides an interpolation method specifically designed for creating hydrologically corrected DEM. Moreover, further spurious sinks have been removed by means of Fill tool. In the scientific literature some methods are known for a more suitable grid

resolution (Hengl, 2006) based on the properties of the input data (i.e complexity of the landsurface), but the grid spacing used seemed suitable for hydro-geomorphological applications since it follows the general rule that it should be adequately sufficient at the local hillslope scale, marking the transition in process dominance from hill slope to channel (Peckam, 2009). This DEM was used in an "object-based" hydro-geomorphological map that was obtained using a step-by-step rule set.

At the first step, a geomorphometric analysis was performed calculating  plan and profile curvatures  at  increasing cell windows: 5, 7, 9, 11, 13, 15, 17, 19 and 21 cells. The multi–scale  analysis of curvatures was performed with Landserf free GIS software, thus obtaining a raster layer for each geomorphometric calculation.

During the second step the best agreement with expert-based geomorphological mapping was achieved with eCognition Developer software by means of an original multiresolution segmentation algorithm, using appropriates land-surface parameters.

The multiresolution segmentation algorithm merges spatially contiguous pixels or cells into "image objects" (segments) based on local homogeneity criteria of the input parameters. These segments, bounded by discontinuities in the input variables, are used further as building blocks in classification, based on attributes such as average values of input variables, shape indexes, and topological relations of segments (Dragut et al., 2013).

More precisely, morphometric parameters obtained in the previous step (plan and profile curvatures at various cell windows) are used with a proportional increased weight to the increasing cell window size for each raster layer (Table 1); as input parameters also sine and cosine of aspect were used. We didn't consider slope gradient because, a part for the valley bottom and hilltop, it is quite constant and didn't give additional information to the segmentation procedure.

Table 1: Weights assigned to each layer implementes in the eCognition developer software for the multiresolution segmentation algorithm.

| Layer (cells window) | Plan curv (5) | Plan curv (7) | Plan curv (9) | Plan curv (11) | Plan curv (13) | Plan curv (15) | Plan curv (17) | Plan curv (19) | Plan curv (21) | Prof curv (5) | Prof curv (7) | Prof curv (9) | Prof curv (11) | Prof curv (13) | Prof curv (15) | Prof curv (17) | Prof curv (19) | Prof curv (21) | Aspect Cos | Aspect Sin |
|---|---|---|---|---|---|---|---|---|---|---|---|---|---|---|---|---|---|---|---|---|
| Weight | 1 | 2 | 3 | 4 | 5 | 6 | 7 | 8 | 9 | 1 | 2 | 3 | 4 | 5 | 6 | 7 | 8 | 9 | 10 | 10 |

Other settings used for this algorithm are: scale 7, shape 0.0002, compactness 0.0002.

During this procedure, the segments obtained were compared to the expert-based geomorphological mapping by using the target-training procedure proposed in Guida et al. (2015) (Fig. 7a).

The image objects derived from the segmentation are shown in Fig. 7b.

In the third step, the classification of the objects, obtained in the previous step, was performed. The classification procedure followed the criteria proposed by Hennrich et al. (1999), whose conceptual background was the 'landscape catena' (Conacher and Dalrymple, 1977), which combines surface form and pedo–hydro–geomorphological processes at hill-slope scale.

In particular, the classification was based on the sum of the planimetric curvatures that were re-classified according to the threshold values listed in the Table 2. The interval values listed in the Table 2 were achieved by a supervised classification.

The use of only the plane curvature sum, computed with different windows, allow to obtain an object-based hydro-geomorphological map (Fig. 7c) with a good accordance with the expert based geomorphological map.

[Figure]

**EXPERT BASED GEOMORPHOLOGICAL MAP (a)**

- Main ridge
- Section ridge
- Hillslope
- Riparian corridor
- Hollow
- Saddle

**SEGMENTATION (b)**

- watershed boundary

Plan curvature sum

- -77.27 - -22.64
- -22.64- -13.36
- -13.36 - -7.78
- -7.78 - -3.71
- -3.71 - -0.90
- -0.90 - 2.30
- 2.30 - 6.12
- 6.12 - 11.61
- 11.61 - 22.12
- 22.12 - 85.31

**HYDRO-GEOMORPHOTYPES (c)**

- watershed boundary
- ridge
- nose
- hillslope
- hollow
- riparian corridor

During the second step the best agreement with expert based geomorphological mapping was achieved with e Cognition software by means of an original multiresolution segmentation algorithm, by assigning a proportional increased weight to the increasing cell window size used for each raster layer (Fig. 7a). The hydro-geomorphological map (Fig. 7b) was obtained by expert based re-classification of the sum of plane curvature classes, choosing threshold values according to the hydrological components (hydro-geomorphotypes) listed in Table 1.

[Figure]

Figure 7. a) Expert- based hydro-geomorphological map; ab) Multi–resolution segmentation map; bc) Object-based hydro-geomorphological map obtained classifying the Multiresolution segmentation map by using only the plan curvature sum.

[revised manuscript text omitted]